# Altered Visceral Adipose Tissue Predictors and Women’s Health: A Unicenter Study

**DOI:** 10.3390/ijerph19095505

**Published:** 2022-05-01

**Authors:** Vanessa Carvalho Moreira, Calliandra Maria de Souza Silva, Izabel Cristina Rodrigues da Silva

**Affiliations:** Graduate Program in Health Sciences and Technologies, Faculty of Ceilandia, University of Brasilia, Federal District, Brasilia 72220-275, Brazil; cdssilva@gmail.com (C.M.d.S.S.); belbiomedica@gmail.com (I.C.R.d.S.)

**Keywords:** obesity, visceral adipose tissue, lipid accumulation product

## Abstract

(1) Background: The excess visceral adipose tissue (VAT) accumulation in women may reflect an early or advanced state of a metabolic disorder and a higher risk of cardiovascular disease than other types of obesity. This study aimed to determine the predictor variables (demographic information, anthropometric data, and blood biomarkers) for changes in VAT in adult women. (2) Methods: This cross-sectional study was conducted with women aged 18–59 years attending nutritional consultation at the Centro Universitário de Brasília (CEUB)’s nutrition school clinic, Brazil. All participants’ medical records were reviewed throughout the study and data of interest were collected. Various anthropometric measurements and biochemical exams were performed and analyzed in a univariate logistic regression model to identify the possible risk factors predictors for the presence of altered VAT. (3) Results: Our logistic regression model considered body mass index (BMI) greater than 25 kg/m^2^, lipid accumulation product (LAP), and waist–hip ratio (WHR) as predictors of altered VAT. (4) Conclusion: LAP has a robust predictive capacity for changes in visceral fat in adult women, followed by WHR and BMI, making these variables effective in assessing the risk for changes in visceral fat and their inclusion essential in the individual and collective clinical practice.

## 1. Introduction

Obesity has become the most frequent metabolic disease in the world, and its prevalence is increasing in all age groups, making it a global public health problem [1,2]. More than 500 million adults worldwide are estimated to be obese, while 1.5 billion are gauged to be overweight, significantly influencing the overall survival of the general population [3,4].

This condition is a complex and multifactorial chronic disease caused by an imbalance between energy intake and expenditure, associated with genetic, environmental, or behavioral issues [5]. In most countries, obesity is prevalent among women [4] and young and middle-aged adults [6] and reflects the importance of weight control in this population group.

Today, one of the most consequential fat distribution patterns is the excessive storage of fat in the abdominal region, known as visceral, abdominal, or central obesity. This type of obesity is marked by an increase in adipose tissue (fat) around the intra-abdominal organs and is associated with the development of cardiometabolic diseases [7]. Women generally have more peripheral and less central fat than men [8], and their body-fat percentage generally increases with age [9]. In this way, excess visceral fat accumulation in women may reflect an early or advanced state of a metabolic disorder [10].

Furthermore, excess visceral adipose tissue (VAT) contributes to a higher risk of cardiovascular disease than other types of obesity [11]. Visceral obesity is considered a risk factor for metabolic syndrome, insulin resistance, dyslipidemia, diabetes mellitus independently of obesity, hypertension, and cardiovascular disease (CVD) [9,12,13].

The body-fat distribution can be assessed by several methods that differ in precision, performance time, and execution cost. Dual-energy X-ray absorptiometry (DXA), magnetic resonance imaging (MRI), and computed tomography (CT) exams are considered the gold standard; however, they are expensive and time-consuming methods, limiting their use in clinical practice and population-based studies [14].

Nonetheless, for use in clinical practice and epidemiological studies, alternative measures can be used to determine visceral adiposity, such as anthropometric measurements, e.g., body mass index (BMI), waist circumference (WC), and waist-to-hip ratio (WHR) [15,16]. Complementarily, measures that integrate anthropometric measurements with lipids serum levels, known as metabolic indices, can also be used, such as the lipid accumulation product (LAP) and the triglycerides and glucose product (TyG), both considered as early markers for insulin resistance [17,18].

Alarmingly, the prevalence of abdominal obesity, followed by excess visceral adipose tissue in women, is increasing [9]. Evidence indicates that anthropometric indicators and serum biochemical analysis are good indicators of obesity, whether general or abdominal, but few studies compare these indicators as predictive factors of changes in visceral fat. Therefore, this study strove to determine the predictor variables (incorporating demographic information, anthropometric data, and blood biomarkers) for changes in visceral fat in low-income adult women (determined by a formula with anthropometric data).

## 2. Materials and Methods

This cross-sectional study was conducted with women aged 18–59 years attending a nutritional consultation at the nutrition school clinic of the Centro Universitário de Brasília (CEUB), Brazil, between January 2016 and July 2018 who fulfilled the study’s eligibility criteria. Medical records with incomplete information, women under 18 and over 60 years, pregnant women, and athletes were excluded from the study.

All patients’ medical records were reviewed throughout the study’s duration, and data of interest were collected. Each participant was evaluated using a standardized anamnesis form to obtain demographic information (including date of birth, gender, income, information on personal diabetes diagnosis, obesity, hypertension, dyslipidemia, or heart disease, as well as information on family history of diabetes, hypertension, or dyslipidemia), anthropometric measurements, and results of serum biochemical tests.

The study did not involve high-risk procedures for the subjects, and all subjects received their test results, individualized food plan, and dietary and lifestyle advice based on the consultation results. The CEUB’s Research Ethics Committee approved this study (CAAE: 76753717.4.0000.0023), and all participants signed the Free and Informed Consent Form.

### 2.1. Anthropometric Parameters

According to the CEUB nutrition school clinic care protocol, the body mass measurements were obtained to the nearest 0.1 kg on a digital scale. Participants were asked to remove shoes and heavy clothing to be weighed, wearing only minimal clothing. For height measurements, the participants remained barefoot in the center of the equipment, with their feet together and arms outstretched at the sides of their body, using a stadiometer with an accuracy of 0.1 cm. Then, the patient’s BMI (body mass index) was calculated and their nutritional status was classified according to the cutoff points recommended by the World Health Organization (WHO) [19].

These anthropometric measurements were taken by trained personnel using standardized procedures [20]. The skinfold thickness was measured on the right side of the body with the aid of a caliper with a precision of 0.1 mm, while the circumference measurements were evaluated with a standard flexible measuring tape with a precision of 1 mm. The body fat percentage (%) was determined by the Jackson and Pollock formula, using seven folds (Jackson and Pollock, 1978; Jackson and Pollock, 1980).

Samouda et al. [21]’s equation and Lee et al. [22]’s equation were used to determine visceral adipose tissue (VAT), muscle mass (MM), and the ratio between them (VAT/MM) (Table 1). The formulas are based on skinfold thicknesses, circumferences of different body regions, body weight, height, and biological sex. The justification for using both equations is their relevance in the literature and their low costs and straightforward measurement parameters. VAT values higher than 105.64 cm^2^ were considered altered (increased).

For Lee et al. [22]’s formula, the circumferences of the limbs (limb C) were corrected by the thickness of the subcutaneous adipose tissue (SAT). The skinfold calibrators (S) measurement was assumed to be twice the SAT thickness. Corrected muscle circumferences in centimeters (including bones) (Cm) were calculated as Cm = limb C − (π S), where limb C is the limb circumference (in cm) and S is the skinfold calibrator measurement. This equation has already been used in similar studies [23,24,25]. To obtain a three-dimensional muscle mass measurement, corrected muscle circumferences were squared and multiplied by the height.

### 2.2. Biochemical Analysis

The laboratory tests results were collected only if the patient’s tests were executed at the CEUB school laboratory within a maximum of 30 days after the nutritional consultation. Serum levels of total cholesterol and fractions, triglycerides, and glucose were recorded. All participants fasted for at least eight hours before laboratory tests. After receiving their results, they were given a nutritional consultation per the protocols established by the laboratory.

The lipid accumulation product (LAP) index was determined using the equation proposed by Kahn [26] for women. For this variable, triglyceride measurements in mg/dL were converted to mmol/L. The TG/HDL ratio was obtained by dividing the triglyceride level (mg/dL) by the HDL-C level (mg/dL). The triglycerides and glucose product (TyG) index was calculated using the logarithm of fasting triglyceride x fasting glucose divided by 2 [27].

The cutoff point and reference values used in this study are described in the Appendix A.

### 2.3. Statistical Analysis

Study variables are described in terms of absolute and relative frequencies (%) or in terms of mean and standard deviations. For identifying possible risk factors for altered VAT, either the chi-square association test or Fisher’s exact test (samples with an expected frequency of less than 5%) was initially applied, in dichotomous tables, with data adjusted according to the cutoff points and separated into “without alterations” and “with alterations” values. Finally, logistic regression was employed to determine the odds ratio for VAT. All data were processed using the Statistical Package for the Social Sciences (SPSS) version 28.0 software, with a significance level of 5%.

## 3. Results

We recruited 163 participants aged between 18 and 59 (mean age was 34.9 ± 11.6 years). The participants’ economic class was consistent with populations served by the public health system since they had lower financial and economic resources levels (income less than 800 dollars). Over half of the participants were overweight (BMI ≥25.0 kg/m^2^; n = 90; 55.21%). Most participants reported not having a personal history of diabetes, hypertension, or dyslipidemia. Regarding family history, most participants reported no diabetes, obesity, dyslipidemia, or heart disease cases within their families. Table 2 presents the sample’s data description. The values obtained from the anthropometric and laboratory measurements variables were organized and described by the mean and standard deviation (Table 2), and their cutoff points presented in the Appendix A were considered for the remaining analyses.

In a first analysis of the data, we adopted the univariate model to identify the possible risk factors for the presence of altered visceral adipose tissue (altered VAT > 105.64 cm^2^; Table 3). A family history of diabetes, obesity, hypertension, or dyslipidemia combined with a personal history of hypertension is considered a possible background for altered VAT. Some anthropometric measurements were also associated with altered VAT, namely: high BMI (body mass index); skinfolds: subscapular and midaxillary mean, large arm, hip, and thigh circumferences; high biochemical variables, such as blood glucose, total cholesterol, triglycerides, HDL, VLDL, and non-HDL; the lipid accumulation product (LAP), TG (triglycerides)/HDL, triglycerides and glucose product (TyG) and TyG-BMI indices; and waist-to-hip and VAT/MM ratios.

However, despite the significant associations with altered VAT in the literature, only the BMI greater than 25 kg/m^2^, LAP, and waist–hip ratio were considered predictors of altered VAT when evaluated in an adjusted logistic regression model combined with other probable variables of altered VAT that considered the Wald test for decision making (Table 4). Therefore, an overweight patient has a 4.5 times higher chance of altered VAT than a patient with normal BMI; in the same line, patients with upper LAP median (26.19 cm·mmol/L) have 18.9 times higher odds of altered VAT, and those with a high waist-to-hip ratio (≥0.8), 6.2 times higher.

## 4. Discussion

Evidence strongly suggests that the factor that mediates the association between obesity and health outcomes is the amount of visceral adipose tissue (VAT) instead of the amount of subcutaneous adipose tissue (SAT) or total body fat [28]. In our study, focusing on a population of multiracial, multi-ethnic adult women from a low-income community, we observed that among the variables analyzed, predictive factors for alteration in VAT were the lipid accumulation product (LAP), waist–hip ratio, and body mass index (BMI).

In our study, the LAP variable had the highest impact on VAT, in which a normal LAP (≤26.19 (cm·mmol/L)) reduced the odds of having an altered VAT (>105.64 cm^2^) at least 18 times. LAP is an indicator proposed by Kahn [26] based on the combination of waist circumference (WC) with fasting triglycerides (TG) serum levels. According to the author, as WC and TG variables tend to accumulate over time, this indicator expresses a constant risk of mortality and cardiovascular alterations in adults. Therefore, the higher the WC and the TG serum levels, the higher the LAP value [26,29].

Studies have used LAP as a central obesity marker to estimate the accumulation and excess of VAT, corroborating this study’s LAP and VAT association [9,16,30]. An elevated waist circumference in the presence of elevated fasting triglycerides is considered a reflection of visceral obesity and is related to several metabolic abnormalities [26,31]. Roriz et al. [13] observed a good LAP accuracy in discriminating visceral fat in adult men and women. In a cohort study, the LAP showed a positive correlation with visceral fat and a negative correlation with leg fat, confirming that the LAP can describe central fat accumulation [32].

An increased LAP value may indicate that several tissues or organs have become more vulnerable to lipid superaccumulation lesions, contributing to their relationship with diverse comorbidities. With advancing age, LAP values tend to be higher in men than women [26].

Elevated LAP levels were associated with type 2 diabetes mellitus and increased mortality from heart failure in women of normal weight and high cardiovascular risk [33]. LAP was also associated with diabetes mellitus incidence [34], metabolic syndrome prediction, and insulin resistance diagnosis [18,31], as well as elevated blood pressure and high serum lipid levels [26,35]. Elevated LAP levels have also been observed in women with polycystic ovarian syndrome (POS), correlating with a significant increase in impaired glucose tolerance prevalence and metabolic disorders caused by the syndrome [32,36]. A Brazilian study associated a high LAP value with an atherogenic profile, with changes in serum levels of total cholesterol, HDL-c, LDL, and apo B, suggesting that LAP can be an easy and straightforward clinical marker for the evaluation of cardiometabolic risk factors [37].

Waist-to-hip ratio (WHR) was another significant predictor variable in this study, in which a normal WHR reduced the odds of having increased visceral fat by up to six times. The WHR is an indicator used to identify the type of body fat distribution, reflecting the relative fat distribution in the upper and lower parts of the body [38,39]. For women, values above 0.8 indicate an android-like fat distribution (also known as a central or abdominal distribution), considered a risk factor for developing cardiovascular diseases when in excess [5].

Studies show the WHR’s importance as an anthropometric indicator for visceral fat accumulation [40,41]. WHR demonstrated good diagnostic concordance with abdominal fat determined by computed tomography (CT), surpassing the results from WC and BMI [42]. According to Gadekar et al. [43], WHR is the best anthropometric measure compared to WC and BMI to predict visceral fat. Conversely, studies have observed a lower WHR correlation with VAT when compared to other anthropometric indicators [44,45,46].

According to Roriz et al. [11], the use of this indicator as a visceral fat predictor should be done cautiously, especially in individuals with variations in weight and adiposity, as their WC measure varies simultaneously with their hip measurement, keeping the WHR constant. Furthermore, the hip measurement does not consider variations in individuals’ pelvic structure nor the reduction of the tissue in this area with the aging process [47]. However, according to Akpinar et al. [48], the WHR is the most useful obesity measure and the best uncomplicated anthropometric index to predict a wide range of risk factors and related health conditions.

Similarly to VAT, high WHR values are associated with chronic non-communicable diseases (such as hypertension, cardiovascular disease, diabetes, and insulin resistance) and mortality [49,50]. Studies have shown that WHR is a better screening measure than other anthropometric indicators for cardiometabolic conditions [51,52]. The waist-to-hip ratio association with increased risk of cardiovascular mortality was observed in cohort studies among women [53,54], in which a higher WHR was related to a higher risk of adverse cardiovascular events and was capable of identifying a higher cardiovascular risk profile in overweight individuals [55]. In contrast, Czernichow et al. [56] found no difference in the WHR’s ability to discriminate the risk of mortality from chronic diseases; likewise, Akpinar et al. [48] found little value in WHR as an indicator of cardiovascular risks for the public.

Central obesity in women, as measured by WHR, was also significantly related to the risk of type II diabetes [57] in obese and non-obese individuals [41]. Similarly, WHR correlated positively with insulin resistance in apparently healthy [8,58], overweight, and obese individuals [59]. Elevated WHR in women has also been associated with a higher risk of liver cirrhosis [60] and reproductive problems [39].

When considering the body mass index (BMI) for obesity classification, our study found that being within the normal range or underweight reduces the odds of having visceral fat above the median by up to five times, while overweight and obesity are risk factors for an increase in VAT values. The BMI, which relates weight to height squared, provides reliable information on excess body weight and is widely used to classify and monitor the overweight and obesity epidemic [48,61], as well as to link obesity status with an increased risk of cardiovascular disease, type 2 diabetes, and mortality [62].

High BMI is associated with an increased risk of cardiovascular disease and type 2 diabetes [48] and is a strong predictor of diabetes in many populations [63]. BMI also performed similarly to bioelectrical impedance analysis (BIA) in predicting body fat distribution [64]. On the other hand, this index has been questioned as a marker of cardiovascular risk since not all obese individuals will develop cardiometabolic complications, and the location of fat accumulation has a significant impact on the manifestation of these comorbidities [65].

Although BMI had a predictive relationship with VAT in the present study, studies have indicated that even if BMI is most commonly used as an indicator of total adiposity, it does not differentiate between muscle and fat, nor does it determine the fat location [26,66]. BMI also presents differences in its values based on age, sex, and ethnicity [67]. Consequently, individuals with a similar BMI can vary considerably in their abdominal fat mass (VAT). Furthermore, BMI does not consider the weight proportion related to increased muscle mass or the body’s excess fat distribution, which affects the health risks associated with obesity [48].

In line with this study’s primary objective, we found that LAP was a better predictor of alteration in VAT measure than other obesity indices, such as BMI and WHR, in women. Although the LAP’s ability to discriminate excess visceral fat is still little explored [11], studies indicate greater specificity of the LAP in the prediction of type 2 diabetes mellitus [32], metabolic syndrome [16], resistance to insulin [12], and risk of cardiovascular diseases [26] when compared to BMI. A Brazilian study found that LAP is a better predictor of metabolic syndrome when comparing the WHR and other anthropometric measures such as BMI [68]. In Bozorgmanesh’s study, LAP was superior to BMI and similar to other adiposity indices, such as WHR, in the discrimination of diabetes.

Studies also suggest that the WHR has greater sensitivity and specificity for assessing the risk of obesity-associated diseases than parameters such as WC and BMI [69,70]. In contrast, an adult Chinese population study reported that WHR and BMI are equally useful indicators for discriminating between individuals with and without metabolic syndrome [71].

The model presented in this study is a good predictive model for changes in visceral fat (predictive value of 88%), and, to the best of our knowledge, it is the first study that evaluates the predictive variables of changes in visceral fat using only clinical anthropometric and laboratory data.

Some limitations of the present study must be taken into account. Firstly, we used information collected from medical records, which might have included errors when originally entered; however, we performed a rigorous data quality assessment to reduce the possibility of information bias. Secondly, the study was cross-sectional, making it impossible to establish causal inferences. Thirdly, the sample included only adult women, so the results cannot be generalized to older women. Fourthly, our study was conducted in a single population care center, so our results cannot be generalized to the population of Brazilian women. Despite these limitations, this study provides evidence of the importance of visceral adiposity in women’s health.

Future studies should include the analysis of men, older adults, and populations of other ethnicities, besides measuring VAT and MM through imaging tests. Given that Brazil is multiracial, working with defined races can change the presented results. Considering that LAP played a prominent role in this study, further studies are needed to confirm LAP’s clinical significance as a predictor of changes in visceral fat.

## 5. Conclusions

In conclusion, considering the importance of visceral fat as one of the determining factors of the metabolic alterations associated with obesity and cardiometabolic diseases, the use of more straightforward, low-cost, and radiation-free methods can be considered an advance in the visceral obesity diagnosis and prevention of outcomes associated with this adiposity. Our results provide evidence that LAP has a robust predictive capacity for changes in visceral fat in adult women, followed by WHR and BMI, making these variables effective in assessing the risk for changes in visceral fat and their inclusion essential in the individual and collective clinical practice.

## Figures and Tables

**Table 1 ijerph-19-05505-t001:** Predictive equations for visceral adipose tissue (VAT) and muscle mass (MM) with the validation results determined by the authors.

Author	Equations	R^2^	SEE
Samouda et al. [21]	VAT(female) = 2.15 × WC − 3.63 × PC + 1.46 × age + 6.22 × BMI − 92,713	0.836	36.88
	VAT(male) = 6 × WC − 4.41 × PC + 1.19 × age − 213.65	0.803	47.73
Lee et al. [22]	MM = Ht × (0.00744 × CAC^2^ + 0.00088 × CTC^2^ + 0.00441 × CCC^2^) + 2.4 × sex − 0.048 × age + race* + 7.8.	0.91	2.2 kg

R^2^: determination coefficient; SEE: standard error of estimate; VAT (cm^2^): visceral adipose tissue; MM (kg): skeletal muscle mass; WC: waist circumference (cm); PC: proximal mid-thigh circumference (cm); BMI: body mass index (kg/m^2^); Ht: height (m); CAC^2^: corrected arm circumference; CTC^2^: corrected mid-thigh circumference; CCC^2^: corrected calf circumference. Age in years; biological sex = 1 for men and 0 for women; race = −1.6 for Asians, 1.2 for African Americans, and 0 for Whites or Hispanics. * In the population we studied, the value 0 was adopted because it was a mixed-race (multiracial) sample, and there was no significant difference when using African-American and Asian races’ constants.

**Table 2 ijerph-19-05505-t002:** Clinical characteristics of the research participants.

Variable		Statistics Summary
		Frequency Yes:No
	Diabetes	3:160
Personal history	Hypertension	16:147
	Dyslipidemia	10:153
	Diabetes	58:105
	Obesity	31:132
Family history	Hypertension	78:85
	Dyslipidemia	30:133
	Cardiovascular disease	31:132
		Mean ± SD
	Height, m	1.61 ± 0.07
	Weight, kg	69.39 ± 15.28
	Subscapular fold, mm	24.72 ± 11.79
	Triceps fold, mm	23.95 ± 11.04
	Mid-axillary fold, mm	20.11 ± 10.57
Anthropometric	Supra iliac fold, mm	22.91 ± 10.12
measures	Chest fold, mm	18.04 ± 10.86
	Abdominal fold, mm	28.03 ± 12.07
	Mid-thigh fold, mm	32.61 ± 12.5
	Calf fold, mm	21.07 ± 11.48
	Arm circumference, cm	30.32 ± 5.13
	Waist circumference, cm	81.85 ± 13.65
	Abdominal circumference, cm	88.77 ± 14.6
	Hip circumference, cm	100.19 ± 14.4
	Calf circumference, cm	37.39 ± 5.68
	Thigh circumference, cm	55.33 ± 8.29
	Waist-to-hip ratio	0.83 ± 0.18
	Jackson and Pollock’s fat percentage (seven-folds), %	30.12 ± 8.33
	Muscular mass, kg	22.51 ± 4.34
	Visceral adipose tissue, cm^2^	99.44 ± 67.72
	VAT/MM ratio cm^2^/kg	4.40 ± 2.81
	Glucose, mg/dL	90.74 ± 14.43
	Total cholesterol, mg/dL	181.66 ± 37.16
	Triglycerides, mg/dL	113.01 ± 78.59
Laboratorial	HDL, mg/dL	56.21 ± 14.7
measures	LDL, mg/dL	103.57 ± 33.89
	VLDL, mg/dL	22.40 ± 15.34
	non-HDL, mg/dL	124.27 ± 39.21
	LAP, cm.mmol/L	35.26 ± 40.56
	TG/HDL	2.32 ± 2.33
	TyG	3.63 ± 0.28
	TyG-BMI	97.49 ± 25.78

BMI = body mass index; HDL = high-density lipoprotein; LDL = low-density lipoprotein; VLDL = very low-density lipoprotein; LAP = lipid accumulation product; TyG = triglycerides and glucose product; TG/HDL = triglyceride to high-density lipoprotein ratio; TYG-BMI = triglyceride glucose product body mass index; VAT/MM = visceral adipose tissue to muscle mass ratio.

**Table 3 ijerph-19-05505-t003:** Distribution of participants according to the increase/alteration of visceral adipose tissue (VAT) associated with the sociodemographic profile and clinical characteristics.

	Visceral Adipose Tissue				*p*-Value	OR (CI OR)
	Altered (n = 70)		Normal (n = 93)			
	Count	%	Count	%		
Personal history (diabetes)	3	100.0%	0	0.0%	0.077	NA
Personal history (hypertension)	12	75.0%	4	25.0%	0.006 *	4.60 (1.42–14.97)
Personal history (dyslipidemia)	5	50.0%	5	50.0%	0.746	1.35 (0.38–4.87)
Family history (diabetes)	34	58.6%	24	41.4%	0.003 *	2.72 (1.40–5.25)
Family history (obesity)	20	64.5%	11	35.5%	0.007 *	2.98 (1.32–6.74)
Family history (hypertension)	44	56.4%	34	43.6%	0.001 *	2.94 (1.54–5.86)
Family history (dyslipidemia)	18	60.0%	12	40.0%	0.037 *	2.34 (1.04–5.25)
Family history (heart disease)	13	41.9%	18	58.1%	0.900	0.95 (0.43–2.10)
Height (m)	24	39.3%	37	60.7%	0.473	0.79 (0.41–1.51)
Weight (kg)	35	53.0%	31	47.0%	0.032 *	2 (1.06–3.78)
BMI (kg/m^2^)	64	71.1%	26	28.9%	<0.001 *	27.49 (10.62–71.18)
Subscapular fold (mm)	41	54.7%	34	45.3%	0.005 *	2.45 (1.30–4.63)
Triceps fold (mm)	27	52.9%	24	47.1%	0.082	1.81 (0.93–3.52)
Mid-axillary fold (mm)	34	54.0%	29	46.0%	0.024 *	2.08 (1.10–3.96)
Supra iliac fold (mm)	29	47.5%	32	52.5%	0.359	1.35 (0.71–2.56)
Chest fold (mm)	25	39.7%	38	60.3%	0.500	0.80 (0.42–1.53)
Abdominal fold (mm)	26	40.6%	38	59.4%	0.630	0.86 (0.45–1.62)
Mid-thigh fold (mm)	25	46.3%	29	53.7%	0.543	1.23 (0.64–2.37)
Calf fold (mm)	32	49.2%	33	50.8%	0.187	1.53 (0.81–2.89)
Arm circumference (cm)	38	55.9%	30	44.1%	0.005 *	2.49 (1.31–4.73)
Waist circumference (cm)	66	78.6%	18	21.4%	<0.001 *	68.75 (22.15–213.42)
Abdominal circumference (cm)	69	58.0%	50	42.0%	<0.001 *	59.34 (7.91–445.43)
Hip circumference (cm)	35	53.8%	30	46.2%	0.022 *	2.10 (1.11–3.98)
Calf circumference (cm)	27	52.9%	24	47.1%	0.082	1.81 (0.93–3.52)
Thigh circumference (cm)	34	56.7%	26	43.3%	0.007 *	2.43 (1.27–4.67)
Glucose (mg/dL)	22	71.0%	9	29.0%	<0.001 *	4.28 (1.82–10.04)
Total cholesterol (mg/dL)	34	54.8%	28	45.2%	0.016 *	2.19 (1.15–4.18)
Triglycerides (mg/dL)	24	72.7%	9	27.3%	<0.001 *	4.87 (2.09–11.35)
HDL (mg/dL)	16	76.2%	5	23.8%	0.001 *	5.22 (1.81–15.05)
LDL (mg/dL)	19	52.8%	17	47.2%	0.177	1.67 (0.79–3.51)
VLDL (mg/dL)	24	75.0%	8	25.0%	<0.001 *	5.54 (2.31–13.32)
non-HDL (mg/dL)	20	64.5%	11	35.5%	0.007 *	2.98 (1.32–6.74)
LAP (cm·mmol/L)	57	80.3%	14	19.7%	<0.001 *	24.74 (10.81–56.64)
TG/HDL	43	63.2%	25	36.8%	<0.001 *	4.33 (2.23–8.42)
TyG	43	60.6%	28	39.4%	<0.001*	3.70 (1.92–7.11)
TyG-BMI	61	77.2%	18	22.8%	<0.001 *	28.24 (11.85–67.31)
Waist-to-hip ratio	57	66.3%	29	33.7%	<0.001 *	9.68 (4.59–20.39)
Jackson and Pollock’s fat percentage (seven-folds) (%)	48	71.6%	19	28.4%	<0.001 *	8.50 (4.16–17.34)
Muscular mass (kg)	32	51.6%	30	48.4%	0.08	1.77 (0.93–3.36)
VAT/MM ratio (cm^2^/kg)	69	90.8%	7	9.2%	<0.001 *	847.71 (101.84–7056.06)

Notes: NA = not applicable; * *p* < 0.05. Abbreviations: BMI = body mass index; CI = confidence interval; HDL = high-density lipoprotein; LDL = low-density lipoprotein; VLDL = very low-density lipoprotein; LAP = lipid accumulation product; OR = odds ratio; TyG = triglyceride and glucose product; TG/HDL = triglyceride to high-density lipoprotein ratio; TyG-BMI = triglyceride glucose product body mass index; VAT/MM = visceral adipose to muscle mass ratio. * The reference values used in this study are available in the Appendix A, and the frequency on each line (each variable) represents the frequency of participants with the altered values.

**Table 4 ijerph-19-05505-t004:** Visceral adipose tissue (VAT) logistic regression model.

		B	E.P.	Wald	gl	Sig.	Exp(B)	95% C.I. to EXP(B)	
								Inferior	Superior
**Step 1 ^a^**	**BMI (** **kg/m^2^)**	**−1.563**	**0.779**	**4.023**	**1**	**0.045**	**0.210**	**0.046**	**0.965**
	Glucose (mg/dL)	−0.365	0.721	0.256	1	0.613	0.694	0.169	2.854
	Triglycerides (mg/dL)	15.559	40192.977	0.000	1	1.000	5716910.805	0.000	.
	VLDL (mg/dL)	−15.905	40192.977	0.000	1	1.000	0.000	0.000	.
	**LAP** **(cm·mmol/L)**	**−2.941**	**1.229**	**5.724**	**1**	**0.017**	**0.053**	**0.005**	**0.588**
	TG/HDL	0.337	0.996	0.114	1	0.735	1.400	0.199	9.870
	TyG	2.297	1.218	3.558	1	0.059	9.943	0.914	108.140
	TyG-BMI	−0.923	0.941	0.960	1	0.327	0.398	0.063	2.516
	**Waist-to-Hip ratio**	**−1.824**	**0.584**	**9.772**	**1**	**0.002**	**0.161**	**0.051**	**0.506**
	Jackson Pollock’s Fat Percentage (seven-folds) (%)	−0.460	0.572	0.647	1	0.421	0.631	0.206	1.937
	Constant	3.375	0.580	33.844	1	0.000	29.225		

Notes: Biological sex = female; ^a^. Variable(s) entered in step 1: BMI classification, glucose (mg/dL), triglycerides (mg/dL), VLDL (mg/dL), LAP, TG/HDL, TyG, TyG-BMI, waist-to-hip ratio, Jackson and Pollock’s fat percentage (seven-fold). In bold: values considered altered for each variable; B = values for the logistic regression equation for predicting the dependent variable from the independent variable; df = degrees of freedom for the model; Sig. = two-tailed *p*-value for Wald chi-square value; Exp(B) = odds ratios for the predictors. Abbreviations: BMI = body mass index; VLDL = very low-density lipoprotein; LAP = lipid accumulation product; TG/HDL = triglyceride to high-density lipoprotein ratio; TyG = triglycerides and glucose product; TyG-BMI = triglyceride glucose product-body mass index. * The reference values used in this study are available in the Appendix A.

## Data Availability

Not applicable.

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
