# Peer review of "Altered Visceral Adipose Tissue Predictors and Women’s Health: A Unicenter Study"

_ijerph, 2022, doi:10.3390/ijerph19095505_

Round 1

Reviewer 1 Report

The goal of the present study was to identify predictors variables of changes in visceral fat in women. The study analyzed anthropometric data and blood biomarkers, and reports a significant association between lipid accumulation product, waist-to-hip-ration, and BMI and increased accumulation of visceral fat in women.

Comments:

  • The purpose of the study should be clearly stated in the abstract.
  • Table 2 should be broken down into two tables: one for sociodemographic description and another for clinical characteristics. As is, the table is very cumbersome to navegate.
  • The manuscript should be revised for proper English grammar.

Author Response

Por favor, o anexo.

Reviewer 2 Report

The authors in their manuscript titled "Altered Visceral adipose tissue predictors and Women’s 2 Health: A Unicenter Study." have looked at various predictive determinants of changes in visceral adipose tissue in this cross-sectional study to determine one that is most accurate. The subject is interesting and manuscript is generally well written. Here are a few of the comments about the manuscript.

1) The written part of the results section needs to be rewritten. It is very cluttered and confusing in this current version and not easy to comprehend.

2) In your opinion how stable a measure is fasting triglyceride which is a part of equation to measure LAP? Can be affected my meals consumed the previous day? Isn't then WHR more accurate to measure VAT.

3) Typos present throughout the manuscript. Please correct.

Reviewer 3 Report

Summary:

In the present paper, the authors present their clinical study results that was carried out in order to demonstrate that certain anthropometric measurements and laboratory data may predict the alterations in visceral fat in adult women and consequently the likelihood of cardiovascular risk.

The hypothesis is interesting, the study is original and the study design is appropriate. The methods are clearly described and the manuscript is well structured. The authors contribute to this field of research with new findings. Nevertheless, some revisions are recommended before publication.

Observations:

Line 46: “according to” should be “in”.

Line 54: “Lipids serum levels” should be “lipid serum levels”.

Line 55: “Like” should be “such as”.

Line 66: ‘The This” should be “This”.

Line 68: “and fulfilled” should be “and having fulfilled”.

Line 92: “measurements were measured” should be “was measured”.

Line 102: “superior to” should be “higher than”.

Line 119: “In our population studied” should be “In the population we studied” or “In the studied population”.

Line 126-27: “were nutritional” should be “and were given nutritional”.

Line 167: the “2” in “kg/m2” should be put into the upper index.

Line 184: “considering” should be “focusing on”.

Line 188: “reduced at least 18 times the chance of having an altered VAT” should be “reduced the odds of having an altered VAT by at least 18 times”.

Line 205: “the advancing age” should be “advancing age”.

Line 208: The scientific data you refer to has very little to do with the paper you cite under the number 31. That paper is about polycystic ovarian syndrome and not cardiovascular diseases. The citation you should insert instead of it is “The Lipid Accumulation Product Is Associated With Increased Mortality in Normal Weight Postmenopausal Women”, https://onlinelibrary.wiley.com/doi/full/10.1038/oby.2011.42

Line 212: Polycistic Ovarian Syndrome should be “POS” instead of “SOP”.

Line 213: “Syndrome” should be “syndrome”.

Line 242: “waist-hip ratio” should be “waist-to-hip ratio”.

Line 246: “amid” should be “in”.

Line 256: “reduces by up to 5 times the chance of having visceral fat above the median” should be “reduces the odds of having visceral fat above the median by up to 5 times”.

Line 296: “which may have errors when filling in” should be “which might have included errors when originally entered”.

Line 303: “analysis in” should be “analysis of”.

Line 305: “Considering” should be “Considering”.

The unit of measurement litre can be either signified by a capital or lowercase l, but try to be consistent about it. Litre seems to be always abbreviated to “l”, but decilitre varies from “dl” to “Dl”. I recommend you choose either the upper or lowercase “l” and stick to it.

Line 145: I’d suggest the number of participants included in the study written right at the start of Results.

Table 2: In the top 8 rows of the table about Personal and Family history, you state that data is presented in “Yes:No (%)” form, but then we can’t see the percentage data, only the “Yes:No” data. I suggest the percentage to be either shown, or when stating the form of data, write it only as “Yes:No” to keep it consistent. The SI symbol of “kilo” is a lowercase “k”, so the kg in the unit of measurement for “Weight”, “Muscle mass” and “VAT/MM ratio” should be written that way. Waist-to-Hip ratio should be written as Waist-to-hip ratio, consistent with the rest of the text outside the tables. This error is present in Table 3 and 4 as well.

Table 3: The SI symbols of litre should be consistent and “Kg” should be written as “kg”. I’m not sure you have to write all the abbreviations down again, such as LAP, HDL or VAT/MM ratio. On the other hand, the meaning of “OR (IC OR)” should be written down.

Table 4: The meaning of “B”, “E.P.”, “Wald”, “gl”, “Sig”, “Exp(B)” and “C.I.” should be written down. Symbol of decilitre should be consistent.

The consistency in the symbols of litre and kilogram should be checked in the supplementary document as well.
